# What Are the Oxidizing Intermediates in the Fenton and Fenton-like Reactions? A Perspective [note 1]

**DOI:** 10.3390/antiox11071368

**Published:** 2022-07-14

**Authors:** Dan Meyerstein

**Affiliations:** 1Chemical Sciences Department, The Radical Research Center and The Schlesinger Family Center for Compact Accelerators, Radiation Sources and Applications, Ariel University, Ariel 4070000, Israel; danm@ariel.ac.il; 2Chemistry Department, Ben-Gurion University, Beer-Sheva 8410501, Israel

**Keywords:** OH^•^, Fe^IV^=O_aq_, CO_3_^−^, pH effect, reactive oxidizing species

## Abstract

The Fenton and Fenton-like reactions are of major importance due to their role as a source of oxidative stress in all living systems and due to their use in advanced oxidation technologies. For many years, there has been a debate whether the reaction of Fe^II^(H_2_O)_6_^2+^ with H_2_O_2_ yields OH^•^ radicals or Fe^IV^=O_aq_. It is now known that this reaction proceeds via the formation of the intermediate complex (H_2_O)_5_Fe^II^(O_2_H)^+^/(H_2_O)_5_Fe^II^(O_2_H_2_)^2+^ that decomposes to form either OH^•^ radicals or Fe^IV^=O_aq_, depending on the pH of the medium. The intermediate complex might also directly oxidize a substrate present in the medium. In the presence of Fe^III^_aq_, the complex Fe^III^(OOH)_aq_ is formed. This complex reacts via Fe^II^(H_2_O)_6_^2+^ + Fe^III^(OOH)_aq_ → Fe^IV^=O_aq_ + Fe^III^_aq_. In the presence of ligands, the process often observed is L_n_(H_2_O)_5−n_Fe^II^(O_2_H) → L^•^^+^ + L_n−1_Fe^III^_aq_. Thus, in the presence of small concentrations of HCO_3_^−^ i.e., in biological systems and in advanced oxidation processes—the oxidizing radical formed is CO_3_^•^^−^. It is evident that, in the presence of other transition metal complexes and/or other ligands, other radicals might be formed. In complexes of the type L_n_(H_2_O)_5−n_M^III/II^(O_2_H^−^), the peroxide might oxidize the ligand L without oxidizing the central cation M. OH^•^ radicals are evidently not often formed in Fenton or Fenton-like reactions.

## 1. General Remarks

In 1894, Mr. Fenton reported that Fe^II^(H_2_O)_6_^2+^ catalyzes the oxidation of tartaric acid by H_2_O_2_ [1]. No mechanism of this process was suggested by Mr. Fenton. Since then, the reaction Fe^II^(H_2_O)_6_^2+^ + H_2_O_2_ has been called the Fenton reaction and the reactions M^n^L_m_ + ROOR’—where M is either Fe or another low-valent transition metal, L is either H_2_O or another ligand, and R and R’ are either H or another substituent—are called Fenton-like reactions.

The Fenton and Fenton-like reactions are of major importance due to two reasons:They are considered to be the major source of oxidative stress in all living systems.They are used in the advanced oxidation technologies/processes that are of major importance in the environmental removal of pollutants.

Due to this prominence, a search in SciFinder for Fenton in 2021 results in 3286 references.

The first mechanisms of the Fenton reaction were suggested in 1932 by two groups in parallel. Bray and Gorin [2] suggested that the mechanism is:Fe^II^(H_2_O)_6_^2+^ + H_2_O_2_ → Fe^IV^=O^2+^_aq_(1)
whereas Haber and Weiss [3,4] suggested that the mechanism of the Fenton reaction is:Fe^II^(H_2_O)_6_^2+^ + H_2_O_2_ → Fe^III^(H_2_O)_6_^3+^ + OH^•^ + OH^−^(2)

The debate whether the oxidizing intermediate formed in the Fenton reaction is Fe^IV^ = O^2+^_aq_ or OH^•^ has lasted for many decades. Thus, even as recently as this year, it has been suggested that reaction (1) is the correct mechanism, at least in neutral solutions [5], and that (2) is the only process even at pH 5 [6].

The difficulty in differentiating between the two mechanisms stems from the fact that both OH^•^ radicals and Fe^IV^=O^2+^_aq_ react with organic substrates, usually by abstracting a hydrogen atom, and often form the same, or similar, radicals. Using EPR to quantify the relative yields of the radicals formed in order to decide whether their sources are OH^•^ radicals often fails due to their different lifetimes [7]. This difficulty was overcome by measuring the final products formed when a mixture of two alcohols is present.^8^ This technique requires that the low-valent metal cation initiating the Fenton-like reaction has a fast ligand exchange rate, i.e., it does not fit Fe^II^(H_2_O)_6_^2+^. Using this technique, it was shown that the reaction Cr^II^(H_2_O)_6_^2+^ + H_2_O_2_ proceeds via a mechanism analogous to reaction (2), whereas the reaction Cu^I^_aq_^+^ + H_2_O_2_ does not yield OH^•^ radicals or Cu^III^_aq_ [8]. Furthermore, thermodynamic arguments [8] and kinetic arguments using the Marcus theory [9] indicate that the Fenton and Fenton-like reactions do not proceed via the outer sphere mechanism. Therefore, an inner sphere mechanism was proposed [8,9]:ML_m_^n+^ + H_2_O_2_ ⇌ {L_m−1_M(H_2_O_2_)^n+^ + L}/{L_m−1_M(HO_2_)^(n−1)+^ + L + H^+^}(3)

For simplicity, it will be assumed in that the complex formed is L_m_M(H_2_O_2_)^n+^. Reaction (3) might be followed by a variety of routes, e.g., [8,9]:→ ML_m_^(n+1)+^ + OH^•^ + OH^−^(4a)
L_m_M(H_2_O_2_)^n+^ → ML_m_^(n+2)+^ + 2OH^−^RH(4b)
→ ML_m_^(n+1)+^ + R^•^ + OH^−^ + H_2_OR=R(4c)
→ ML_m_^(n+1)+^ + HOR-R^•^ + OH^−^(4d)

Naturally, L_m__−1_M(H_2_O_2_)^n+^ might also directly oxidize different substrates, e.g., inorganic reducing agents.

It was later discovered that when the central cation M has a too high redox potential, e.g., Co(II) [10], or cannot be oxidized, e.g.: Al^III^, Ga^III^, In^III^, Sc^III^, Y^III^, La^III^, Be^II^, Zn^II^, and Cd^II^ [11,12,13], the binding of two or more peroxides to the central cation might lead to the formation of OH^•^ radicals via disproportionation of the peroxides without involving oxidation of the central cation [10,11,12,13]:M^n^_aq_ + kH_2_O_2_ ⇌ M^n^(HO_2_^−^)_k−1_(H_2_O_2_)_aq_ + (k−1)H^+^ (k = 2 or 3)(5)
M^n^(HO_2_^−^)_k-1_(H_2_O_2_)_aq_ → M^n^(HO_2_^•^)(HO_2_^−^)_k−2_(OH^−^)_aq_ + OH^•^(6)

The observation that ligated H_2_O_2_ can oxidize a second ligated peroxide suggests that it might also oxidize other ligands. This was tested theoretically, by DFT [14], and experimentally for the oxidation of a carbonate ligated to Co^II^ [15], thus proving this possibility.

## 2. The Fenton Reaction Is (Fe(H_2_O)_6_^2+^ + H_2_O_2_)

Efforts to determine whether the reaction Fe(H_2_O)_6_^2+^ + H_2_O_2_ forms OH^•^ radicals via following the formation of the DMPO-OH^•^ adduct by EPR failed, as it was shown that even mild oxidants, e.g., Fe^III^_aq_, oxidize DMPO via [16]:DMPO + Ox → DMPO^•^^+^ + Red(7)
DMPO^•^^+^ + H_2_O → DMPOH^•^ + OH^−^(8)

The rate constant of the Fenton reaction in acidic media is *k*(Fe(H_2_O)_6_^2+^ + H_2_O_2_)~50 M^−1^s^−1^. The measured rate constants depend on the pH and on the ratio [H_2_O_2_]/[Fe(H_2_O)_6_^2+^]; the latter dependencies mainly stem from the observation that in the presence of excess H_2_O_2_ reactions (9) [17] and (10) [17,18] contribute to the observed rate constants [17].
Fe^III^_aq_ + H_2_O_2_ ⇌ Fe^III^(HO_2_) + H^+^ (*k*_9_ = 69 M^−1^s^−1^ *k*_−9_ = 0.11 s^−1^ at pH 2.0) (9)
Fe(H_2_O)_6_^2+^ + Fe^III^(HO_2_) → Fe^III^_aq_ + {Fe^III^_aq_ + OH^•^}/{Fe^IV^=O_aq_}(10)
*K*_10_ = 7.7 · 10^5^ M^−1^s^−1^ at pH 1.0

The nature of the products of reaction (10) were later determined [19] to be Fe^III^_aq_ + Fe^IV^=O_aq_; thus, clearly in acidic solutions when [H_2_O_2_]/[Fe(H_2_O)_6_^2+^] > 1, a mixture of OH^•^ radicals and Fe^IV^=O_aq_ is formed.

Next, Bakac et al. developed a new procedure to differentiate between OH^•^ radicals and Fe^IV^=O_aq_ based on the different final products formed in the reactions of OH^•^ radicals and Fe^IV^=O_aq_ with DMSO, (CH_3_)_2_SO [20]. This technique can only be used for iron. Using this technique, it was proved that, in acidic solutions, OH^•^ radicals are formed by the Fenton reaction, whereas in neutral solutions, where pH > 6, the product is Fe^IV^=O_aq_ [20]. This proves that the Fenton reaction under physiological conditions does not form OH^•^ radicals: However, this statement is not correct for the acidic organelles, e.g., lysosomes [21] and some peroxisomes [22]. This conclusion is correct for reactions of Fe(H_2_O)_6_^2+^, but not for all Fenton-like reactions of Fe^II^L_m_, as seen below.

Recently, it was shown that the Fenton reaction is dramatically accelerated in the presence of low concentrations of bicarbonate well below those present in living cells [19]. The oxidizing transient formed under these conditions is the carbonate anion radical, CO_3_^•^^−^ [19]. CO_3_^•^^−^ is a strong oxidizing agent, E^0^(CO_3_^•^^−^/CO_3_^2^^−^) =1.57 V vs. NHE [23] and is evidently somewhat stronger in neutral media. CO_3_^•^^−^ is still a considerably weaker oxidizing agent than OH^•^ radicals and is, therefore, more selective as a ROS [24,25]. The reactions occurring were proposed to be [19]:Fe(H_2_O)_6_^2+^ + H_2_O_2_ ⇌ (H_2_O)_5_Fe(O_2_H)^+^/(H_2_O)_3_Fe(O_2_H)^+^ + H_3_O^+^(11)
(H_2_O)_5_Fe(O_2_H)^+^/(H_2_O)_3_Fe(O_2_H)^+^ + HCO_3_^−^ → Fe^III^_aq_ + CO_3_^•^^−^(12)
Fe(H_2_O)_6_^2+^ + HCO_3_^−^ ⇌ (H_2_O)_3_Fe(CO_3_) + H_3_O^+^ + 2H_2_O(11a)
(H_2_O)_3_Fe(CO_3_) + H_2_O_2_ → Fe^III^_aq_ + CO_3_^•^^−^
(12a)
Recent unpublished results [26] suggest that reaction (12) likely proceeds via:(H_2_O)_5_Fe(O_2_H)^+^/(H_2_O)_3_Fe(O_2_H)^+^ + HCO_3_^−^ → (CO_3_)Fe^IV^_aq_(13)
and reaction (12a) likely proceeds via:(H_2_O)_3_Fe(CO_3_) + H_2_O_2_ → (CO_3_)Fe^IV^_aq_(13a)

The (CO_3_)Fe^IV^_aq_ thus formed might decompose via:(CO_3_)Fe^IV^_aq_→ (14)
→ Fe^III^_aq_ + CO_3_^•^^−^(14a)

Substrate
→ Fe^III^_aq_ + oxidized-substrate + HCO_3_^−^(14b)

The competition between reactions (14a) and (14b) depends on the substrate. Thus, for DMSO *k*_14a_ >> *k*_14b_, but for PMSO (phenyl-methyl-sulfoxide) *k*_14a_~*k*_14b_.

## 3. Fenton-like Reactions Involving Fe^II^L_m_

Two types of Fenton-like reactions have to be considered.

When ligands, L, different from H_2_O are ligated to The Fe^II^ central cation, the effect of HCO_3_^−^ on the mechanism, discussed above, can be included herein. It should be noted that the technique to distinguish between OH^•^ radicals and Fe^IV^=O_aq_, developed by Bakac et al. [20], cannot always be applied here because the mechanism of the reaction LFe^IV^=O with DMSO is not known. The mechanism of the reactions of Fe^II^L_m_ with H_2_O_2_ for the following ligands was studied.

L = PO_4_^3^^−^/HPO_4_^2^^−^ [20]. The results suggest that the Fenton reaction in the presence of phosphate in neutral solutions yields OH^•^ radicals and not (PO_4_^3^^−^)_m_Fe^IV^=O_aq_ [20].L = edta [22]. The reaction Fe^II^(edta)^2^^−^ + H_2_O_2_ was studied at pH > 5.5 using the technique developed by Masarwa et al. [8]. The results indicate that OH^•^ radicals are the product of this reaction [27].L = nta, nta = N(CH_2_CO_2_^−^)_3_^3^^−^ [28]. The reaction Fe^II^(nta)^−^ + H_2_O_2_ was studied. Surprisingly, though edta and nta are very similar ligands, the results differ considerably. The results suggest that the major product of the Fe^II^(nta)^−^ + H_2_O_2_ is a (nta)Fe^IV^=O_aq_ complex [28]. The yields of the final products are pH dependent [28].L = citrate [29]. The reaction of Fe^II^(citrate)^−^ with H_2_O_2_ was studied. This reaction is of importance because Fe^III^(citrate) is a major component of the non-transferrin iron mobile pool [30]. The results indicate that the reaction Fe^II^(citrate)^−^ + H_2_O_2_ in neutral solutions does not yield OH^•^ radicals. The results do not answer the question whether a Fe^IV^(citrate)_aq_ species is a transient formed by this reaction. When low concentration of HCO_3_^−^ are added to this system, the kinetics and final products are changed dramatically, indicating that the CO_3_^•^^−^ radical anion is a major product of the reaction under these conditions [29].

The results presented in this section indicate that the mechanism of the Fenton-like reactions of Fe^II^L_m_ complex dramatically depend on the nature of the ligand. Therefore, one cannot assume that Fe^II^ complexes with analogous ligands react via the same mechanism.

When different peroxides are used as oxidants in the Fenton-like reaction, such as in biological systems, the most important peroxides are the ROOH compounds, where R is an alkyl. The ROOH peroxides are formed in biological systems, mainly in lipids, via the chain reaction [30,31]:RH + Ox → R^•^ + Ox-H/(Ox^−^ + H^+^) (Ox = OH^•^, R’^•^, Fe^IV^=O_aq_ etc.)(15)
R^•^ + O_2_^•^ → RO_2_(16)
RH + RO_2_^•^ → RO_2_H + R^•^(17)

Therefore, the mechanism of the reaction (CH_3_)_3_COOH + Fe(H_2_O)_6_^2+^ was studied. The results indicate that in this system Fe^IV^=O_aq_ is also formed in neutral solutions in the absence of bicarbonate. In the presence of low concentrations of bicarbonate, CO_3_^•^^−^ radical anions are the product of this Fenton-like reaction [32].

The S_2_O_8_^2−^ and HSO_5_^−^ peroxides are of major importance in advanced oxidation technologies [33,34,35,36]. Therefore, the mechanisms of the reactions Fe(H_2_O)_6_^2+^ + HSO_5_^−^/S_2_O_8_^2−^ were studied. The results indicate that in acidic media, SO_4_^•^^−^ radical anions are the active oxidizing species formed, in neutral solutions, Fe^IV^=O_aq_ is formed, and in the presence of low concentrations of bicarbonate, CO_3_^•^^−^ is the oxidizing intermediate formed [26].

## 4. Other Fenton-like Reactions

Fenton-like reactions are reported for most low-valent transition metals and even for cations that are not involved in redox processes [11,12,13]. Herein, only Fenton-like reactions involving Cu^I^ [37] and Zn^II^ [38,39,40,41] that are of biological importance and Co^II^, due to its role in advanced oxidation technologies [15], are discussed.

The reaction of Cu^I^ with H_2_O_2_ was long thought to yield OH^•^ radicals [42], but it was later shown that the active oxidizing agent is Cu^I^(H_2_O_2_) [8] or Cu^III^_aq_ [43]. It was also proposed that the reaction of Cu^I^ with S_2_O_8_^2−^ yields Cu^III^_aq_ [44]. Conversely, it was proposed that the reactions of Cu(II) with HSO_5_^−^and S_2_O_8_^2−^ yield Cu^III^_aq_ and SO_4_^•^^−^ [45].

Surprisingly, Zn^2+^_aq_ and Zn^II^-complexes were shown to be involved in the formation of reactive oxygen species (see references [38,39,40,41] for example.). However, no chemical mechanism initiating this process was forwarded. One possible mechanism is that suggested by Shul’pin et al. [13]. According to this mechanism, the reactions involved are:Zn^2+^_aq_ + H_2_O_2_ ⇌ Zn^II^(O_2_H^−^)^+^_aq_ + H^+^(18)
Zn^II^(O_2_H^−^)^+^_aq_ + H_2_O_2_ ⇌ Zn^II^(O_2_H^−^)(H_2_O_2_) ^+^_aq_(19)
Zn^II^(O_2_H^−^)(H_2_O_2_)^+^_aq_ → Zn^2+^_aq_ + OH^•^ + HO_2_^•^ + OH^−^(20)

As the steady state concentration of H_2_O_2_ in biological media is very low, the probability that two H_2_O_2_ will bind to the same Zn^2+^_aq_ is low. Therefore, it is tempting to propose that the process leading to the formation of reactive oxygen species catalyzed by Zn^2+^_aq_ is:Zn^2+^_aq_ + HCO_3_^−^ ⇌ Zn^II^(HCO_3_^−^)^+^_aq_(21)
Zn^II^(HCO_3_^−^)^+^_aq_ + H_2_O_2_ ⇌ Zn^II^(HCO_3_^−^)(H_2_O_2_)^+^_aq_(22)
Zn^II^(HCO_3_^−^)(H_2_O_2_)^+^_aq_ → Zn^2+^_aq_ + OH^•^ + CO_3_^•^^−^ + H_2_O(23)

These two plausible mechanisms must be studied experimentally to prove one or both of them.

The reaction Co(H_2_O)_6_^2+^ + H_2_O_2_ to yield OH^•^ radicals is endothermic due to the high redox potential of the Co^III/II^ couple [10]. However, it was shown that the following reactions replace the simple Fenton-like reaction [14]:Co(H_2_O)_6_^2+^ + 3H_2_O_2_ ⇌ (H_2_O)Co^II^(HO_2_^−^)_2_(H_2_O_2_)(24)
(H_2_O)Co^II^(HO_2_^−^)_2_(H_2_O_2_) → (H_2_O)Co^II^(HO_2_^−^)(HO_2_^•^)(OH^−^) + OH^•^(25)

In the presence of bicarbonate, the complex *cyclic-*(CO_4_)Co^II^(HO_2_^−^)_2_(H_2_O) is formed. This complex decomposes via [15]:*cyclic-*(CO_4_)Co^II^(HO_2_^−^)_2_(H_2_O) → (H_2_O)Co^II^(HO_2_^•^)(OH^−^)_2_ + CO_3_^•^^−^(26)

The reaction of HSO_5_^−^ with Co(H_2_O)_6_^2+^ and with Co^II^(P_2_O_7_)(H_2_O)_2_^2−^ require more than one peroxymonosulfate to form radicals [46].

Finally, it should be pointed out that it is likely that ligands other than carbonate, with the proper redox potential, might also be oxidized directly by peroxides [14].

## 5. Heterogeneous Fenton-like Processes

A variety of heterogeneous catalysts react with H_2_O_2_ in Fenton-like processes. Thus, ZnO-nanoparticles induce the formation of reactive oxygen species in biological systems. However, this is attributed to the dissolved Zn^2+^_aq_ ions [39] and is, therefore, not truly heterogeneous.

The most important heterogeneous catalysts of Fenton-like processes have iron atoms/cations as the active participants, e.g., zero-valent iron [47], MFe_2_O_4_ (e.g., Fe_3_O_4_ [48] and MgFe_2_O_4_ [49]), and LaFeO_3_ [50]. These systems are used in advanced oxidation processes and not in biological ones. Therefore, their mechanisms are not discussed herein.

## 6. Concluding Remarks

The major conclusions of this perspective are:The reaction Fe^II^(H_2_O)_6_^2+^ + H_2_O_2_ yields OH^•^ radicals as the active oxidizing agent in acidic solutions when [Fe^II^(H_2_O)_6_^2+^] > [H_2_O_2_], a mixture of OH^•^ radicals and Fe^IV^=O_aq_ in acidic solutions when [Fe^II^(H_2_O)_6_^2+^] < [H_2_O_2_], Fe^IV^=O_aq_ in neutral solutions, and CO_3_^•^^−^ in solutions containing even low concentration of HCO_3_^−^, i.e., under physiological conditions.It is important to note that mechanisms of the reactions H_2_O_2_ + Fe^II^L_m_(H_2_O)_k_, where L are ligands different than water, depend dramatically on the properties of L. Thus, one must study the mechanism for each ligand separately.The study of the mechanisms of Fenton-like reactions with other peroxides requires separate studies.The mechanisms of Fenton-like reactions of other low-valent metal cations differ from each other and thus require separate studies.

Therefore, it must be concluded that the mechanism of each Fenton-like reaction should be studied before concluding which oxidizing transient is formed in that reaction.

## Data Availability

Not applicable.

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
