# Peer review of "What Are the Oxidizing Intermediates in the Fenton and Fenton-like Reactions? A Perspectiveâ€"

_antioxidants, 2022, doi:10.3390/antiox11071368_

Round 1
Reviewer 1 Report
The short Review by D. Meyerstein gives a quick overview of the important Fenton reaction in the classic conditions and in some variants. It reviews a good amount of recent and historic literature on the subject. The organization is generally good.
The problem in the manuscript lies in the formatting. Either it has been hastily done or some copy/paste errors were made with the template. As a consequence several sentences are confusing.
Below, I point out several sentences that need to be revised. Throughout the text, several instances of extra spaces and similar typos can be found.
Line 36 is not followed by reaction 1, which is bumped together with reacton 2.
Line 55: the meaning is not clear at all.
Line 55/56: following/follows is repeated 3 times in two rows, consequently the sentence is less readable.
Line 74: reaction (9) and (10) are mentioned, but they should probably be (8) and (9).
Line 82: comma use is confusing.
Line 89: the charge/radical signs seem to be subscripts (it may be a problem of the font, however).
Line 94: The is capitalized in the middle of the sentence.
Line 119: the paragraph name/number should be 2. not I.
Line134/136: the verb is missing.
Bibliography: several references are not correctly reformatted.
Reviewer 2 Report
The review is well written and could be published in the present form.
1. There are several typos in the text: Line 70 capital O in Oxidise; Lines 102-105 edta and nta should be capital; Line 155 double "the"; and some others.
2. Section "Other Fenton Like reactions" could be supplemented with information about Fenton Like reactions involving Zn ions as they are also of biological importance.
